# Cardioprotective and Hepatoprotective Potential of Silymarin in Paracetamol-Induced Oxidative Stress

**DOI:** 10.3390/pharmaceutics16040520

**Published:** 2024-04-09

**Authors:** Bogdan Okiljević, Nikola Martić, Srđan Govedarica, Bojana Andrejić Višnjić, Milana Bosanac, Jovan Baljak, Branimir Pavlić, Isidora Milanović, Aleksandar Rašković

**Affiliations:** 1Department of Cardiac Surgery, Dedinje Cardiovascular Institute, 11000 Belgrade, Serbia; bogdanokiljevic@gmail.com; 2Department of Pharmacology, Toxicology, and Clinical Pharmacology, Faculty of Medicine, University of Novi Sad, 21000 Novi Sad, Serbia; aleksandar.raskovic@mf.uns.ac.rs; 3Clinic of Urology, Clinical Center of Vojvodina, 21000 Novi Sad, Serbia; 902004d22@mf.uns.ac.rs; 4Faculty of Medicine, University of Novi Sad, 21000 Novi Sad, Serbia; 904007d23@uns.ac.rs; 5Department of Histology and Embryology, Faculty of Medicine, University of Novi Sad, 21000 Novi Sad, Serbia; bojana.andrejic-visnjic@mf.uns.ac.rs (B.A.V.); milana.bosanac@mf.uns.ac.rs (M.B.); 6Faculty of Technology, University of Novi Sad, 21000 Novi Sad, Serbia; bpavlic@uns.ac.rs; 7Department of Pharmacology, Biochemistry, Pharmacy and Ecology, Academy for Applied Studies Belgrade, College of Health Sciences, 11080 Belgrade, Serbia; isidora.milanovic@assb.edu.rs

**Keywords:** silymarin, paracetamol, mice, antioxidant activity, hepatoprotective, cardioprotective

## Abstract

Silymarin, derived from *Silybum marianum*, has been used in traditional medicine for various ailments. In this study, the cardioprotective and hepatoprotective effects of silymarin against paracetamol-induced oxidative stress were examined in 28 male Swiss Webster mice, divided into four groups and treated for 7 days (via the oral route) with (a) saline 1 mL/kg (control group), (b) saline 1 mL/kg + single dose of paracetamol 110 mg/kg on the 7th day; (c) silymarin 50 mg/kg; and (d) silymarin 50 mg/kg + single dose of paracetamol 110 mg/kg on the 7th day. In vitro and in vivo antioxidant activity together with liver enzyme activity were evaluated. Histopathological and immunohistochemical assessment was performed. Silymarin mitigated paracetamol-induced liver injury by reducing oxidative stress markers such as lipid peroxidation and restoring antioxidant enzyme activity. Silymarin treatment resulted in a significant decrease in liver enzyme levels. Reduced necrosis and inflammatory infiltrate in liver tissues of silymarin-treated groups were detected as well. Immunohistochemical analysis demonstrated reduced expression of inflammatory markers (COX2, iNOS) and oxidative stress marker (SOD2) in the liver tissues of the silymarin-treated groups. Similar trends were observed in cardiac tissue. These results suggest that silymarin exerts potent hepatoprotective and cardioprotective effects against paracetamol-induced oxidative stress, making it a promising therapeutic agent for liver and heart diseases associated with oxidative damage.

## 1. Introduction

*Silybum marianum*, also referred to as milk thistle and belonging to the Asteraceae family, is one of the oldest and most thoroughly studied herbs from antiquity for liver and gallbladder disorder treatment, including cirrhosis, jaundice, and hepatitis, as well as defense against poisoning from the *Amanita phalloides* mushroom and other toxins [1]. Since its preparations became legally available for therapeutic use in 1969, its usage has expanded widely throughout Europe [2].

Silymarin, a standardized extract made from the seeds of *S. marianum*, is the plant’s active ingredient. It contains between 70% and 80% of the silymarin flavonolignans and 20% to 30% of a chemically undefined fraction, primarily composed of polymeric and oxidized polyphenolic compounds [3]. Silibinin, with a concentration of 60–70%, emerges as the primary and most active component of silymarin [4].

The main chemical distinction between silymarin and other flavonoids lies in the substitution of a coniferyl alcohol group into silymarin’s isomers. Among the three isomers constituting silymarin, silibinin stands out as the most active. Under the brand names Legalon™ or Hepatron™, silymarin is listed in the pharmacopeia of many countries. It is frequently utilized as supportive therapy for chronic liver diseases such as steatosis and alcohol-related liver disease as well as food poisoning caused by fungus [5,6]. According to the Medicines and Medical Devices Agency of Serbia, it is registered under the trademarks Favora^®^ and Carsil^®^.

The imbalance between the production of reactive oxygen species (ROS) and antioxidant defenses, known as oxidative stress, can result in tissue damage [7]. Superoxide radicals, hydroxyl radicals, singlet oxygen, and hydrogen peroxide are highly unstable and reactive oxygen species (free radicals) that are formed by the partial reduction of oxygen [8].

Various tissues and organ systems, such as the liver, kidney, cardiovascular, and nervous systems, are implicated in drug-induced oxidative stress as a mechanism of toxicity [9]. The liver, a major target of ROS attack, experiences disrupted homeostasis when ROS levels become excessive, leading to oxidative stress—a significant factor in the development of chronic and degenerative diseases such as liver disease [10].

Due to its extremely high metabolic rate and highest production rate of reactive oxygen species, specifically hydrogen peroxide (H_2_O_2_) per gram of tissue, injuries brought on by oxidative stress will affect the heart. Furthermore, compared to other organs, the heart possesses lower quantities of antioxidants and overall antioxidant enzyme activity. Despite its high basal rate of reactive species production, the heart exhibits numerous antioxidant defense deficiencies when compared to organs like the liver or kidney [11]. Studies performed on isolated perfused hearts demonstrate that even brief exposure to oxygen radicals diminishes high-energy phosphate levels, impairs contractile ability, and results in structural anomalies [8].

One of the most well-known and widely used models in pharmacology and toxicology research when evaluating the antioxidant activity of herbal therapeutics is paracetamol (APAP, acetaminophen)-induced oxidative stress [12]. Widely used analgesic and antipyretic medication available over the counter [13] is considered to be safe at therapeutic doses, but when taken in excess, it can cause harmful side effects [14]. Nephrotoxicity, ex-trahepatic lesions, hepatic necrosis, and even death in humans and experimental animals are caused by lipid, DNA, and protein peroxidation, significant reductions in hepatic GSH (reduced glutathione) levels, changes in the antioxidant enzyme system, a reduction in the activity of hepatic δ-aminolevulinic acid dehydratase (δ-ALA-D), and an increase in the production of various inflammatory cytokines [13].

Since the cardioprotective effect of silymarin was tested mainly in chemotherapeutic drug-induced [15,16,17] or ischemia-induced [18] cardiotoxicity, the aim of this study was to evaluate the cardioprotective effect of silymarin in paracetamol-induced oxidative stress in mice. The hepatoprotective effect was studied as well.

## 2. Materials and Methods

### 2.1. Chemicals

Silymarin and paracetamol were acquired from Sigma-Aldrich, St. Louis, MO, USA. Based on a Certificate of Analysis by Sigma Aldrich, the silymarin (Product Number: S0292, Batch No. BCCD8696) used in this study contains 46.5% of silibilin. From Sigma-Aldrich (Steinheim, Germany), reagents were supplied as follows: Trolox, gallic acid, 2,2-diphenyl-1-picrylhydrazyl (DPPH), 2,4,6-tris(2-pyridyl)-s-triazine (≥99.0%) (TPTZ). 2,2′-Azino-bis (3-ethylbenzothiazoline-6-sulfonic acid) diammoniumsalt (98%) was purchased from J&K, Scientific Ltd. (Beijing, China). Ferro sulphate heptahydrate and ferric chloride hexahydrate were supplied from Centrohem (Stara Pazova, Serbia).

### 2.2. Animals

The Swiss Webster strain of male, mature, and white laboratory mice was provided by the Military Medical Academy of the University of Defense in Belgrade. Mice were kept and bred, with initial average body weight of 30 ± 3 g, in Ehret Uni-Protect cabinets with a High-Efficiency Particulate Air (HEPA) filter system (EHRET Labor- und Pharmatechnik GmbH & Co. KG, Emmendingen, Germany) in the vivarium of the Institute of Pharmacology, Toxicology, and Clinical Pharmacology of the Faculty of Medicine, University of Novi Sad. The animals were maintained in polycarbonate transparent cages with a regular 12 h day/night cycle, controlled temperature (22–24 °C), and air humidity (55 ± 1.5%). The animals had unlimited access to pellet food and water throughout the experiment, except at the end of the seven-day treatment, when the animals were fasted for 12 h before and 6 h after receiving a toxic unidose of paracetamol (110 mg/kg, p.o.). Animal care and handling adhered to protocols and national guidelines, with all experimental procedures involving animals approved by the Ethical Commission for the Protection of Animal Welfare of the University of Novi Sad with approval number 04-81/10, and by the Ministry of Agriculture, Forestry, and Water Management—Veterinary Administration, with approval number 323-07-07211/2020-05. Ethical guidelines provided by the EU Directive 2010/63/EU on animal welfare and under the Law of Animal Welfare of the Republic of Serbia (OG RS 41/09) were followed in all experimental methods and animal care. The dose of test compounds adapted to mice was calculated from the standard human dose of 70 kg using the formula for converting between human and animal doses.

### 2.3. In Vivo Experimental Design

A total of 28 mice were divided randomly into 4 groups of 7 each. A trained individual administered each treatment using a gastric tube designed for mice at the same time each day over a seven-day period. The groups were distributed as follows:First (control) group—saline solution 1 mL/kg, seven days p.o.;Second group—saline solution 1 mL/kg seven days p.o. + toxic unidose of paracetamol 110 mg/kg p.o.;Third group—silymarin 50 mg/kg, seven days p.o.;Fourth group—silymarin 50 mg/kg, seven days p.o. + toxic unidose of paracetamol 110 mg/kg p.o.

Paracetamol was administered to the animals as a hepatotoxic agent. Silymarin was stored at −20 °C in a freezer, protected from light, and dissolved in physiological solution before daily use. Paracetamol was stored at room temperature, dissolved in saline solution before treatment, heated at 60 °C with simultaneous stirring on a magnetic stirrer, and then cooled to 37 °C immediately before administration. Sacrifice was performed by decapitation 24 h after administration of the toxic dose of paracetamol, followed by complete autopsy and sampling of the liver, heart, and whole blood tissues.

### 2.4. Antioxidant Activity

#### 2.4.1. DPPH Assay

Following the Brand-Williams et al. method, the 2,2-diphenyl-1-picrylhydrazyl (DPPH•) free radical scavenging activity of silymarin was evaluated [19]. The DPPH methanolic solution was previously prepared in a concentration of 26 mg/L and dilution of the reagent was adjusted with methanol to produce an absorbance of 0.70 (±0.02). In the 10 mL test tube, 2.9 mL of DPPH reagent was combined with 0.1 mL of appropriately diluted silymarin (0.40 mg/mL) and then incubated for 60 min at room temperature. Absorbance was measured in three replicates at 517 nm (6300 Spectrophotometer, Jenway, Stone, UK). Using freshly prepared Trolox aqueous solutions (0–0.8 mM, R^2^ = 0.999) and measuring its free radical scavenging, the calibration curve was established. The results were presented as mM of Trolox equivalents per g of silymarin.

#### 2.4.2. FRAP Assay

In accordance with an assay previously published by Benzie and Strain, the reducing power of extracts targeting Fe^3+^ was determined [20]. In order to freshly prepare the FRAP reagent, 300 mM acetate buffer (pH = 3.6), 10 mM 2,4,6-tris (2-pyridyl)-s triazine (TPZT) made up in 40 mM HCL solution, and 20 mM FeCl_3_ aqueous solution were mixed in 10:1:1 (*v*/*v*/*v*) ratio. FRAP reagent in a volume of 2.9 mL and 0.1 mL solution of silymarin (0.1 mg/mL) was mixed and then incubated in the dark for 10 min at 37 °C. Absorbance measuring was performed at 593 nm in three replicates (6300 Spectrophotometer, Jenway, Stone, UK). Using freshly prepared Fe^2+^ (Fe_2_SO_4_) aqueous solutions (0–0.23 mM, R^2^ = 0.999), calibration was conducted. The results were presented as mM of Fe^2+^ equivalents per g of silymarin.

#### 2.4.3. ABTS Assay

A method earlier reported by Re et al., with added modifications, was followed to measure the scavenging capacity of extracts targeting ABTS free radicals [21]. Preparing a mixture of 7 mM ABTS (2,20-Azino-bis (3-ethylbenzothiazoline-6-sulfonic acid) diammonium salt) and aqueous solutions of 2.45 mM K_2_S_2_O_8_ in a 1:1 (*v*/*v*) ratio, an ABTS stock solution was freshly made and incubated in the dark for 16 h. To obtain an absorbance of 0.70 (±0.02), 300 mM of acetate buffer (pH = 3.6) was diluted with the prepared ABTS stock solution. ABTS reagent (2.9 mL) was mixed with 0.1 mL of properly diluted silymarin, and afterward, it was incubated in the dark at ambient temperature for 5 h. Absorbance measuring was performed at 734 nm in three replicates (6300 Spectrophotometer, Jenway, Stone, UK). Using freshly prepared Trolox aqueous solutions (0–0.8 mM, R^2^ = 0.999) and measuring its free radical scavenging, the calibration curve was established. The results were presented as mM of Trolox equivalents per g of silymarin.

### 2.5. Liver Function Tests

Aspartate aminotransferase (AST, EC 2.6.1.1), alanine aminotransferase (ALT, EC 2.6.1.2) creatinine, uric acid, direct bilirubin, concentrations of urea, and their enzymatic activity were evaluated from the serum samples utilizing the automatic analyzer AU480 (Beckman Coulter Inc., Indianapolis, IN, USA). Biochemical studies were conducted using spectrophotometric methods with commercially available kits and following the provided instruction manuals [22].

### 2.6. Histopathology and Immunohistochemistry Assessment

Histological evaluation was blinded and performed by light microscopy by two researchers independently. Liver and heart tissues were sampled in small pieces in order to perform histological analysis, after which they were fixed in formalin solution for 24 h. Samples were embedded in paraffin blocks after being dehydrated in a graded series of isopropyl alcohol. Using a rotation microtome (Sakura Finetek USA, Inc., Torrance, CA, USA), four successive 5 µm thick tissue sections were cut for each animal. Hematoxylin and eosin (H&E) method was used for routine staining. According to the guidelines provided by the manufacturer, the three remaining samples underwent immunohistochemistry staining, when COX2 (ab283574, Abcam, Cambridge, UK), iNOS (ab283655, Abcam, Cambridge, UK), and SOD2 (P04179, Cusabio, Houston, TX, USA) antibodies were utilized. The tissue slides were analyzed under a microscope (Leica DMLB 100T) and photographed with a camera (Leica MC 190 HD) at a magnification of 200×, 400×, and 630×. COX2-, iNOS-, and SOD2-stained tissue slides underwent a semiquantitative evaluation of the intensity of immunopositivity using a four-level scale (grades from 0 to 3). The immunopositivity was assessed as strong (3), moderate (2), weak (1), and no immunopositivity (0). The presence of necrosis and inflammatory infiltrate in the liver tissue was assessed on H&E-stained tissue sections and graded semiquantitatively (0—necrosis/inflammatory infiltrates are not present, 1—necrosis/inflammatory infiltrate are present). Percentages of the tissue surface showing the expression of COX2, iNOS, and SOD2 were also semiquantitatively evaluated. The following histopathological changes were observed on heart tissue sections: interstitial edema; cardiomyocyte edema; cardiomyocyte disorganization; vacuoles; necrosis; disorganization of myofilaments; nucleus appearance changes; the presence of neutrophils.

### 2.7. Determination of In Vivo Antioxidant Activity

To determine the liver’s oxidative status, the lipid peroxidation (LP) levels, and the activities of oxidative stress enzymes such as glutathione peroxidase (GPx), glutathione reductase (GR), total superoxide dismutase (T-SOD), catalase (CAT), glutathione reductase (GR), and glutathione S-transferase (GST), were measured in the liver homogenates. Spectrophotometric procedures were used for all measurements. At 4 °C, the liver tissues were homogenized with a solution of TRIS-HCl buffered in a ratio of 1:3 (*w*/*v*). Those liver homogenates were used to assess oxidative stress indices and lipid peroxidation. For every sample, measurements were conducted twice. The quantity of malondialdehyde (MDA), a byproduct of lipid breakdown brought on by peroxidation damage, was utilized to indirectly evaluate the intensity of LP [23]. The reaction between xanthine and xanthine oxidase, which leads to the formation of a superoxide anion radical and the reduction of the oxidized cytochrome c, is applied to measure the specific activity of T-SOD. At 550 nm, the decrease rate is measured spectrophotometrically [24]. The rate of H_2_O_2_ decomposition at 240 nm was used to measure CAT activity [25]. GR and GPx activities were established by measuring the reduction in absorbance brought on by NADPH oxidation at 340 nm [26,27]. GST activity was determined by the conjugation of 1-chloro-2,4-dinitrobenzene (CDNB) with the glutathione thiol group, measured at an absorbance maximum of 340 nm [26].

### 2.8. Statistical Analysis

Data comparison between animal groups was performed using Student’s *t*-test and one-way analysis of variance (ANOVA) followed by Tukey’s post hoc test. The data are shown as mean ± standard deviation (SD).

## 3. Results

### 3.1. In Vitro Antioxidant Potential Examinations

According to Khatri et al., *S. marianum* and its major bioactive compounds, silymarin, have shown promising results in terms of radical scavenging activity towards DPPH, ABTS^+,^ and superoxide anion radicals, as well as good reducing power towards Fe^3+^ and Cu^2+^ ions determined by the FRAP and Cuprac method, respectively [28]. In order to determine the actual antioxidant capacity, it is necessary to apply tests based on different mechanisms. The antioxidant activity of silymarin was determined by a combination of different in vitro tests (DPPH, FRAP, and ABTS). The ABTS and DPPH tests were used to determine the ability to capture radicals, while the FRAP test determines the reducing power. The results are shown in Table 1, expressed as Trolox equivalents, that is, as the concentration of a Trolox solution (mM) with an antioxidant capacity equivalent to that determined for 1.0 mM of the tested sample [29]. According to Villegas-Aguilar et al., the obtained in vitro antioxidant activity of *S. marianum* extracts determined by FRAP and ABTS tests was 1.4 ± 0.3 mM Fe^2+^/g and 1.3 ± 0.1 mM TE/g of extract, respectively. The lower results observed in the *S. marianum* extracts could be explained by the decrease in bioactivity due to the co-extraction of other compounds with less antioxidant capacity compared to silymarin [30].

### 3.2. Effects of Silymarin on Liver and Kidney Function Tests, Lipid Profile, and Oxidative Stress Enzyme Measurments

Administration of a toxic unidose of paracetamol led to a statistically significant increase in the activity of transaminases (AST, ALT) in serum compared to the control (*p* < 0.05). The ALT and AST activity was statistically significantly lower in the group that was treated with silymarin before paracetamol administration compared to the group that was treated with saline before paracetamol administration (*p* < 0.05). The results are presented in Table 2. In animals treated with saline and silymarin, the activity of AST and ALT was higher compared to the control group, but this difference is not statistically significant. There is no significant difference in total bilirubin concentrations in the serum of animals treated with saline and paracetamol compared to the group treated with a combination of silymarin and paracetamol. Serum urea, creatinine, and uric acid concentrations, which serve as markers of renal function, were also measured. There were no statistically significant differences in urea and creatinine concentrations between the control group, the group treated with saline and paracetamol, silymarin, and silymarin in combination with paracetamol. The uric acid concentration in the serum of animals treated with saline and paracetamol was significantly lower compared to the other three groups, *p* < 0.05. The concentration of triglycerides in the serum of animals treated with saline and a toxic unidose of paracetamol was statistically significantly higher compared to the control group, *p* < 0.05. The group of animals that were treated with saline and silymarin had a significantly lower triglyceride concentration in comparison to the group treated with saline and paracetamol, *p* < 0.05. Comparing the concentrations of cholesterol, HDL, and LDL, no statistically significant differences were registered.

Administration of a toxic single dose of paracetamol in the group treated with saline led to a statistically significant increase in the concentration of malondialdehyde (MDA) and an increase in the intensity of lipid peroxidation in the liver homogenate compared to the Con S group, which confirms that hepatocyte cell membranes were damaged (Table 3). The concentration of MDA, as a biomarker of the intensity of lipid peroxidation in the liver homogenate, was statistically significantly lower in the group treated with silymarin before the toxic dose of paracetamol compared to the group treated with saline and paracetamol. In the group of mice treated with silymarin and silymarin and paracetamol, there was no statistically significant difference between the concentration of malondialdehyde (MDA) in the liver homogenate compared to the Con S group. In the group of mice treated with saline and paracetamol, there was a statistically significant decrease in the specific activity of catalase (CAT), glutathione peroxidase (GPx), superoxide dismutase (SOD), and glutathione-S-transferase (GST) in the liver homogenate compared to the Con S group. There was also a decrease in the specific glutathione reductase (GR) activity, but without statistical significance. The specific activities of the enzymes catalase (CAT), glutathione peroxidase (GPx), superoxide dismutase (SOD), and glutathione-S-transferase (GST) in the homogenate of the liver of mice were statistically significantly higher in the group that received silymarin treatment before the toxic unidose of paracetamol (S50+P), compared to the Con P group. Antioxidative enzyme activities in mice liver homogenate did not show statistically differences in the groups S50 and S50+P compared to the Con S group.

The concentration of MDA, as a biomarker of the intensity of lipid peroxidation in the heart homogenate, was significantly lower in the group treated with silymarin before the toxic dose of paracetamol compared to the Con P group, which confirms that silymarin had protective effects on heart muscle cells. In comparison with the group treated with a combination of saline and paracetamol, the MDA concentration was significantly lower in the group treated with saline and silymarin. In the S50 group, there was no statistically significant difference between the concentration of malondialdehyde (MDA) in the heart homogenate compared to the Con S group (Table 4). Administration of paracetamol in the group treated with a saline solution led to a statistically significant decrease only in the specific activity of catalase (CAT) and superoxide dismutase (SOD), while it led to a decrease in the activity of glutathione peroxidase (GPx), glutathione reductase (GR), and glutathione-S-transferase (GST) in the heart homogenate, but without statistical significance compared to the Con S. The specific activity of catalase (CAT) and superoxide dismutase (SOD) in the heart homogenate was statistically significantly higher in the group that was treated with silymarin before the toxic unidose of paracetamol, compared to the group that was treated with saline and paracetamol (Con P). Also, there was an increase in the specific activity of glutathione peroxidase (GPx), glutathione reductase (GR), and glutathione-S-transferase (GST), but without statistical significance. In the group of mice treated with saline and paracetamol, silymarin, and a combination of silymarin and paracetamol, there were no statistically significant differences in the activity of GPx, GR, and GST in the heart homogenate compared to the Con S group.

### 3.3. Histopathological Analysis of Liver Tissue

Treatment with saline solution and paracetamol (Con P group) led to more pronounced necrosis and inflammatory infiltrate, which was statistically significant compared to the liver tissue of mice treated with saline (Con S) and saline and silymarin (S50). Application of silymarin before the toxic dose of paracetamol (S50+P) significantly reduced the presence of necrosis and inflammatory infiltrate compared to the Con P group (*p* < 0.05) (Table 5, Figure 1).

Administration of a toxic dose of paracetamol in the Con P group led to a statistically significant rise in COX2 staining of a moderate and strong intensity. The liver of mice in the group treated with silymarin before the toxic dose of paracetamol exhibited a statistically significant reduction in moderate COX2 expression compared to the Con P group (*p* < 0.05) (Figure 2).

Similar to the COX2 expression, administration of a toxic dose of paracetamol in the Con P group caused iNOS expression in more samples, along with a statistically significant rise in moderate and strong iNOS expression compared to the Con S group. Administration of silymarin previous to paracetamol (S50+P) led to statistically significantly reduced liver damage compared to the Con P group (*p* < 0.05). Treatment with silymarin and co-treatment with silymarin and paracetamol did not show statistically significant differences compared to the Con S group (Figure 3).

The toxic dose of paracetamol (Con P) led to a statistically significant rise in the frequency and intensity of SOD2 staining. The Con P group had significantly higher staining of moderate and strong intensity compared to the Con S group. SOD2 expression in the liver tissue of mice treated just with silymarin (S50) does not differ from the livers of mice with the saline treatment (Con S). Silymarin administered before a toxic dose of paracetamol (S50+P) led to statistically significantly reduced SOD2 staining intensity compared to the liver tissue of mice receiving only paracetamol (Con P) (*p* < 0.05), although it was higher compared to the saline-only treatment (Con S) (*p* < 0.05). Treatment with silymarin only (S50) did not show statistically significant differences in SOD2 expression compared to the Con S group (Figure 4).

### 3.4. Histopathological Analysis of Heart Tissue

On HE-stained heart tissue sections of the Con P group, morphological alterations were visible in the form of vacuolization, necrosis, cardiomyocyte disorganization, and nucleus appearance changes. Such pathological changes were not detected in the heart tissue of the Con S and S50 group. The S50+P group heart tissue specimens exhibited some of the morphological changes such as vacuolization and oedema, but even those were to a lesser extent compared to the Con P group (Figure 5).

A toxic dose of paracetamol in the Con P group, compared to the Con S group, led to a statistically significant rise in COX2 expression, particularly manifested by a rise in staining of moderate and strong intensity. This overexpression due to paracetamol was statistically significantly (*p* < 0.05) reduced by application of silymarin before paracetamol, so in the S50+P group, a weak or moderate intensity staining was mostly observed, while a strong intensity was not manifested. COX2 expression in the S50 and S50+P groups was not statistically different compared to the Con S group (Figure 6).

iNos expression in the Con S group was recorded in only 30% of the samples, and only at a low intensity. After the administration of paracetamol (Con P), the iNOS expression was significantly more pronounced compared to Con S (*p* < 0.05)—it was observed in more than 80% of the Con P-stained liver tissue, in the highest percentage of moderate intensity, and in some of them strong. When silymarin was co-administered with paracetamol, iNOS expression was significantly reduced compared to Con P (*p* < 0.05). S50+P expression, manifested mainly by a weak staining intensity, resembled the Con S group, with no statistical differences detected among these groups. Although in the S50 group a moderate and even strong iNOS expression were detected in a small number of cases, these differences were not statistically significant compared to the Con S group (Figure 7).

Administration of a toxic dose of paracetamol in the Con P group caused a statistically significant rise in SOD2 expression, from less than 20% of weak staining (Con S) to nearly 80% (Con P), of which more than half was moderate and strong expression. Co-treatment with silymarin and paracetamol led to a statistically significant decreased SOD expression compared to Con P (*p* < 0.05). In S50+P, strong immunopositivity was annulled, and compared to Con S, there were no statistically significant differences (Figure 8).

The extensivity of COX2, iNOS, and SOD2 expression, measured through the percentage of heart tissue showing positive staining of any intensity, rose significantly after the administration of a toxic dose of paracetamol (Con P group) compared to the Con S group. Silymarin treatment before paracetamol led to a statistically significant reduction in extensivity compared to the Con P group (*p* < 0.05). In the silymarin-only treated mice, a slight increase was noted compared to the Con S; however, this was not statistically significant (Table 6).

## 4. Discussion

The mechanisms of action of silymarin are not yet sufficiently comprehended. The available literature suggests that silymarin and silibinin have four distinct mechanisms of action. Firstly, they function as antioxidants, scavengers, and regulators of intracellular glutathione levels. Secondly, they act as stabilizers of cell membranes and regulators of permeability, preventing the entry of hepatotoxic agents into liver cells. Thirdly, they promote the synthesis of ribosomal RNA, which stimulates liver regeneration. Lastly, they inhibit the transformation of stellate hepatocytes into myofibroblasts, a process that contributes to the deposition of collagen and the development of liver cirrhosis. [5].

The key mechanism providing hepatoprotection is the removal of ROS by silymarin. Anti-inflammatory and anti-cancer properties have also been reported. Pharmacokinetic studies have shown that silymarin, when ingested orally, is absorbed and distributed in the digestive tract (liver, stomach, intestines, pancreas). It is most often excreted in the form of metabolites via bile and undergoes enterohepatic recirculation [5].

Based on pharmacological investigations, silymarin has been established as a safe herbal product. Animal studies have shown that silymarin is nontoxic. Other studies demonstrated that silymarin is not teratogenic and no postmortem toxicity has been detected. Adverse effects have been reported including headache, dermatological symptoms, and gastroenteritis as the most common [31]. Silymarin interferes with the expression of cell cycle regulators and proteins involved in apoptosis, thus modulating the balance between cell survival and apoptosis [32]. Paracetamol-induced toxicity is one of the main causes of poisoning worldwide. In the case of overdose, paracetamol causes fatal hepatic necrosis and liver failure, induces oxidative stress such as peroxidation of lipids, DNA, and proteins, significantly reduces the level of glutathione in the liver, and leads to changes in the system of antioxidant enzymes, resulting in a decrease in the activity of hepatic dehydratase, d-aminolevulinic acid, and increases in various inflammatory cytokines. Although nephrotoxicity is less expected than hepatotoxicity in overdose, renal tubular damage and acute renal failure can occur even in the absence of liver injury, leading to death in humans and experimental animals [13]. Various studies in animals such as rabbits, rats, and mice have proven that oxidative stress plays a key role in the toxic effects caused by paracetamol. The significance of mitochondrial damage in paracetamol-induced liver damage has been examined, along with the oxidative stress caused by mitochondrial injury [13]. Induction of acute liver injury by paracetamol is one of the most commonly used experimental models of acute liver injury in mice. The major toxic metabolites found in mice are the same as those found in humans, making the pathogenesis of liver damage in mice directly comparable to that in humans. Specific values of this model are the highly reproducible, dose-dependent hepatotoxicity of paracetamol, as paracetamol overdose is one of the most common causes of acute liver failure in humans [33].

Compared to the liver, cardiac tissue has a higher rate of oxidative metabolism and reduced antioxidant defense, making it particularly vulnerable to damage from ROS [34]. In addition to its hepatoprotective properties, silymarin has been reported to be protective against oxidative stress and myocardial infarction caused by ischemia-reperfusion in rat cardiac tissues. Silymarin possesses cardioprotective effects through many mechanisms, including anti-inflammatory, improved antioxidant defense systems, free radical scavenging, membrane stabilizing, iron-chelating activity, and inhibition of apoptosis [35]. When toxic doses of paracetamol are administered, N-acetyl-p-benzoquinone imine (NAPQI) has a harmful impact on the heart. An overdose of paracetamol can lead to a significant increase in the concentration of NAPQI in the myocardium, which can cause oxidative stress, inflammation, and damage to heart tissue. The elevated concentrations of NAPQI in the heart primarily arise from overwhelming the liver with an excessive dosage, rather than being a consequence of cardiac metabolism [36]. Mechanisms involved in paracetamol-induced hepatotoxicity can also be considered for cardiotoxicity. Sulfhydryl groups of glutathione play a role in the vasodilator response, through endogenous vascular relaxing factor (EDRF). NAPQI depletes the glutathione stores of myocytes, enabling the degradation of EDRF and thus imposing an adverse impact on the myocardium [36].

### 4.1. Effects of Silymarin on Paracetamol-Induced Oxidative Stress—Serum Biochemical Parameters

In this study, the hepatoprotective and nephroprotective effects of silymarin, as well as its effect on the lipid status in mice exposed to oxidative stress using toxic doses of paracetamol, were investigated. The administered dose of paracetamol caused hepatotoxicity and consequently a marked disruption of biochemical parameters and indicators of liver function in the serum of experimental animals. The administered dose of paracetamol (110 mg/kg) led to a significant increase in ALT and AST enzymes in the serum of animals treated with saline and a toxic unidose of paracetamol compared to the control group, which indicates consequent hepatocellular damage and hepatotoxicity caused by a toxic unidose of paracetamol. The results obtained in this study are in agreement with similar papers in the literature [37,38,39,40]. Bektur et al. carried out research regarding the hepatoprotective properties of silymarin in a mouse model in which a toxic dose of paracetamol was also administered. The study findings revealed a significant reduction in aminotransferase activity when animals were pretreated with silymarin, as compared to the group that received only paracetamol [41]. The results of our study are in accordance with these results and indicate a potential hepatoprotective effect of silymarin in conditions of oxidative stress induced by paracetamol. In these experimental conditions, specifically, the dosage of paracetamol used was not sufficient to produce substantial harm to the kidneys in all parameters, except for the uric acid concentration. The study partially demonstrated the potential of silymarin to protect the kidneys, while also suggesting that paracetamol may not be highly selective in causing kidney damage [13]. The concentration of triglycerides in the group treated with silymarin and paracetamol is lower compared to the group treated with paracetamol, which calls into question the effect of silymarin on this parameter, considering that the effect of a toxic dose of paracetamol on the concentration of triglycerides has been recorded. This allows for more investigation focused on that specific aspect of the potential impact of silymarin. The effect of silymarin on paracetamol-induced hepatotoxicity and nephrotoxicity in rats by Gopi et al. serum triglycerides was analyzed. It was noted that the level of triglycerides in animals treated with paracetamol and silymarin was significantly lower compared to the group treated with paracetamol [42]. The data we obtained align with these findings and demonstrate that silymarin has an impact on blood triglyceride levels.

In our study, the administration of silymarin (S50+P) for a duration of 7 days effectively prevented the adverse consequences caused by a toxic dose of paracetamol. This was evident by the levels of MDA, which were comparable to the control group and significantly lower than those observed in the group treated with saline solution (Con P). The intensity of lipid peroxidation in the group treated with silymarin (S50) was not statistically significantly different from the control group (Con), which further confirmed the safety of silymarin administration. In the research conducted by Papackova et al., the hepatoprotective effect of silymarin on liver damage in mice due to the administration of a toxic dose of paracetamol was examined. The treatment with silymarin at a dose of 100 mg/kg lasted for 4 days, with the animals receiving a toxic dose of paracetamol (300 mg/kg) 2 h after the last silymarin treatment. After administration of a toxic dose of paracetamol, the animals were sacrificed after 6 h, 12 h, and 24 h. The study found that the timing of sacrifice significantly affects the evaluation of lipid peroxidation. Specifically, it was shown that only after 24 h following the administration of a hazardous dose of paracetamol was there an increase in MDA in the liver homogenate compared to the control group. Pretreatment with silymarin displayed a propensity to reduce the concentration of MDA in comparison to the negative control, aligning with the findings of our study [43]. In the results obtained by Taghiabadi et al., the cardioprotective effect of silymarin was confirmed in conditions of acrolein-induced cardiotoxicity and oxidative stress in mice. An intraperitoneal dose of 50 mg/kg of silymarin was administered for 7 days, followed by daily oral silymarin administration with acrolein (7.5 mg/kg) through a gastric tube for 2 weeks. Treatment with silymarin and acrolein in the group resulted in a significant decrease in MDA values in the heart homogenate compared to the group treated only with acrolein, which aligns with the findings of our study [44].

### 4.2. Effects of Silymarin on Paracetamol-Induced Oxidative Stress—Liver Tissue

The results obtained in our study indicated that there was a statistically significant decrease in the specific activity of the antioxidant protection enzymes CAT, GSHPx, SOD, and GST in the liver homogenate of the group treated with saline solution and a toxic unidose of paracetamol in comparison to the control group. Pretreatment with silymarin for 7 days prevented the effects of a toxic unidose of paracetamol that was administered, whereby the specific activity of the enzyme was similar to that of the control group and statistically significantly higher compared to the group treated with saline solution and a toxic unidose of paracetamol. Results in accordance with our study were reported by Simeonov et al. in a group of rats that received pretreatment with silymarin at a dose of 100 mg/kg for 7 days, after administration of a toxic dose of paracetamol (600 mg/kg i.p.), where a statistically significant increase in the specific activity of CAT, GPx, and GST enzymes in the liver homogenate was obtained [45]. Hamza et al. showed that treatment of mice with silymarin (50 mg/kg) and paracetamol (p.o.) for 30 days showed a statistically significant increase in the specific activity of CAT and GPx enzymes in the liver homogenate, compared to the group treated with saline and paracetamol. The results of this study are in accordance with ours [39]. In addition to the aforementioned analyses, the toxic effect of paracetamol is supported by histopathological changes in liver tissue (necrosis and infiltration by inflammatory cells), statistically significantly more present in the Con P animals compared to the untreated Con S group. While silymarin treatment alone did not cause histological impairment, co-treatment with silymarin and paracetamol alleviated tissue damage to a level statistically insignificant compared to the Con S, control group. The same or similar changes were noted in earlier studies [46,47,48]. As in our study, Pawar et al. found the same hepatoprotective effects on paracetamol-induced liver damage if administered 7 days before paracetamol treatment [48].

### 4.3. Effects of Silymarin on Paracetamol-Induced Oxidative Stress—Heart Tissue

This study partially confirmed the potential cardioprotective effect of silymarin, based on its antioxidant effect under conditions of induced oxidative stress. In addition to the ability to capture free radicals, silymarin has shown its protective effect on some antioxidant protection enzymes (catalase and superoxide dismutase). Treatment with silymarin (S50+P) for 7 days prevented the decrease in the specific activity of CAT and SOD enzymes, due to the administration of a toxic unidose of paracetamol. Administration of a toxic unidose of paracetamol did not result in a statistically noteworthy decrease in antioxidant protection enzymes, with the exception of CAT and SOD. In the work conducted by Mokhtar et al., it was shown that the reduced activity of SOD can be attributed to the inhibition due to an excessive amount of hydrogen peroxide, and the decrease in catalase activity can be attributed to the inhibition due to the large number of superoxide radicals, which are produced from the mitochondria during an overdose of paracetamol [49]. The results indicate that the model of induction of oxidative stress by paracetamol is not suitable enough to cause strong oxidative stress in the heart, unlike some other potentially cardiotoxic drugs (doxorubicin, cisplatin). In the conditions of cardiotoxicity caused by the use of cisplatin, the use of silymarin prevented the reduction in the specific activity of SOD, while the other enzymes were not shown [31]. In conditions of induced myocardial infarction due to ischemia, pretreatment of rats with silymarin for 7 days prevented the reduction in the specific activity of CAT and SOD, which is in accordance with our results [18].

It is important to say that there are no histological signs of heart tissue damage that are specific and exclusive for paracetamol-induced damage. While earlier studies observed only a few of those non-specific signs of tissue damage, we performed a thorough analysis, which included follow-up of parameters of heart tissue damage (interstitial edema; cardiomyocyte edema; cardiomyocyte disorganization; vacuoles; necrosis; disorganization of myofilaments; nucleus appearance changes; the presence of neutrophils). The presence of vacuolization, necrosis, cardiomyocyte disorganization, and nucleus appearance changes in Con P animals prove the paracetamol-induced cardiotoxicity. Although the previous studies observed that histopathological changes were not specific, the toxic effect was proven by the disruption of cardiomyocyte contractility, which consequently leads to impairment of heart function [36,50,51]. However, in order to obtain a final diagnosis of cardiotoxicity and impairment of heart function, it is necessary to perform an electrocardiography—ECG—and echocardiography—ECHO [52,53]. Also, earlier cases of toxic effects of paracetamol on the heart were accompanied by hepatotoxicity [50]. The same results were obtained in our study, where the toxic effect of paracetamol on liver tissue was proven through morphological signs of tissue damage, as well through the altered intensity and extensivity of COX2, iNOS, and SOD2 expression. The use of paracetamol in the Con P group led to a stronger and more widespread expression of COX2 in the heart and liver tissue. Such results were not expected considering that the prevailing opinion in the literature is that paracetamol inhibits prostaglandin synthesis in vivo and selectively inhibits COX2 [54]. COX2 is considered an inducible form of cyclooxygenase, with a significant role in the development of inflammation, but the cytoprotective effects achieved by COX1 and COX2 via prostaglandins should not be overlooked in cases of hepatotoxicity caused by numerous drugs, including paracetamol [55]. Using pharmacological analyses and genetic tests, it was established that COX, and especially COX2, represents an endogenous protective response to drug-induced liver damage. This would adequately explain the increase in COX2 expression in the tissue of the Con P group, which is consistent with the findings of other authors [56]. Accordingly, the results of reduced expression of pro-inflammatory COX2 in the S50+P group, in which the animals were pretreated with silymarin before the toxic dose of paracetamol, are not surprising. This finding suggests that silymarin alleviated hepatotoxicity, as manifested by a decrease in COX2 expression. Similar results were obtained by Eltahir et al. on the liver in CCl4-induced hepatotoxicity [57]. Paracetamol hepatotoxicity is marked by substantial levels of oxidative stress. Nevertheless, the origin, pathophysiological function, and possibility for targeted therapy of this entity have been presented with inconsistent interpretations. Prior research supported the notion that cytochrome P450 produces ROS during the process of paracetamol metabolism, leading to extensive lipid peroxidation and subsequent liver damage. Nevertheless, later research effectively refuted this assumption, and the prevailing viewpoint now indicates that mitochondria are the primary origin of ROS, which hinder mitochondrial activity and play a role in cell signaling that leads to cell demise. Recently, mitochondria have been increasingly acknowledged as the primary origin of oxidative stress following an excessive intake of paracetamol [58]. Since the cardiomyocytes are extremely rich in mitochondria, a rise in iNOS expression in the heart tissue, as well as in the liver, was expected after paracetamol exposure, confirming the oxidative stress damage in the tissues [59]. A decrease in the intensity and extensivity of iNOS staining after silymarin pretreatment is a confirmation of its cardio- and hepatoprotective potential through its previously demonstrated antioxidative capacity [60,61].

Like in our study, Guo et al. reported that after paracetamol overdose, MDA, a lipid peroxidation marker, was increased, mitochondrial oxidative stress and dysfunction occurred, cellular GSH was depleted, and SOD activity was also significantly decreased in the animal liver tissue after paracetamol exposure [62]. Compared to those literature data, our results suggesting an increase in SOD2 immunoexpression in the liver and heart tissue could appear erroneous or at least doubtful. However, one should bear in mind that spectrophotometrically determined tissue SOD activity includes both isoforms: MnSOD (SOD2), located in mitochondria and Cu/Zn SOD, which is found in cytosol. These isoforms, besides subcellular location, differ in other features too. Mladenovic et al. noted that the activity of MnSOD increased within the first 6 h after applying paracetamol. This increase was a result of hepatocytes adapting to the excessive production of ROS caused by paracetamol. Given that mitochondria are significant cellular producers of reactive oxygen species, it is unsurprising that the activity of mitochondrial SOD in cardiomyocytes is elevated. Unlike MnSOD, the activity of cytosolic Cu/Zn SOD remained consistently reduced over the initial 48 h following paracetamol intoxication [38]. In our study, paracetamol overdose induced SOD2, both in intensity and extensivity, proving that mitochondria are significantly affected in hepatocytes and cardiomyocytes. A reduction in both parameters of SOD2 expression in hepatocytes and cardiomyocytes speaks in favor of silymarin protective features, but the exact mechanism of its action is yet to be elucidated. Collectively, this indicates that focusing on reducing oxidative stress in the mitochondria could be a potentially effective treatment strategy for paracetamol overdose in a clinical setting. One of the ways that N-acetylcysteine, the sole antidote now available for paracetamol overdose patients, protects is by neutralizing ROS and decreasing oxidative stress in the liver. Additional potent antioxidants, particularly those targeted at mitochondria, should be thoroughly examined in clinical trials, provided their safety in humans is confirmed [58].

## 5. Conclusions

This study confirmed that the well-known hepatoprotective effects of silymarin could be extended and have a beneficial effect on indicators of cardiac muscle function. Seven-day use of silymarin in laboratory mice led to a decrease in the level of liver transaminases in the serum of those groups of animals treated with a toxic dose of paracetamol. Administration of silymarin also had a favorable effect on indicators of kidney function as well as lipid status in laboratory animals. The beneficial effect of silymarin led to the return of malondialdehyde levels and concentrations of oxidative stress enzymes in the liver and heart homogenates to the levels before paracetamol application. It was shown on histological preparations that the use of silymarin before the toxic dose of paracetamol could significantly reduce the presence of necrosis and inflammatory infiltrate. Bearing all this in mind, it would be justified to conduct additional studies on animals that would confirm the role of silymarin in preventing damage not only to the liver but also to heart tissue caused by toxic drug metabolites or xenobiotics.

## Figures and Tables

**Figure 1 pharmaceutics-16-00520-f001:**
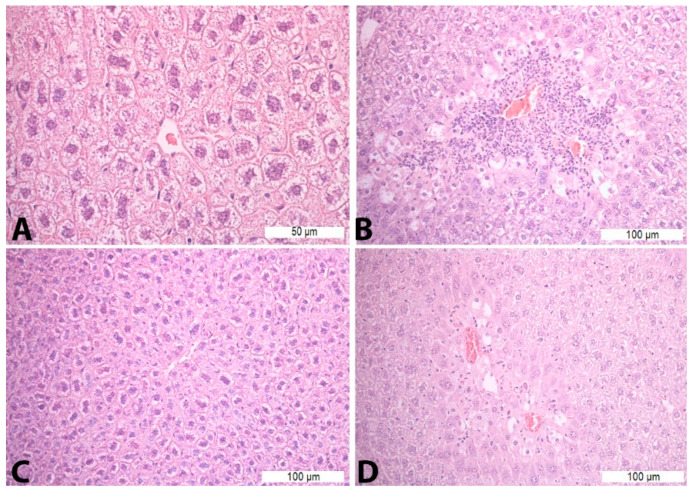
Histopathological analysis of the liver tissue—H&E staining: (**A**)—Con S (400×); (**B**)—Con P (200×); (**C**)—S50 (200×); (**D**)—S50+P (200×).

**Figure 2 pharmaceutics-16-00520-f002:**
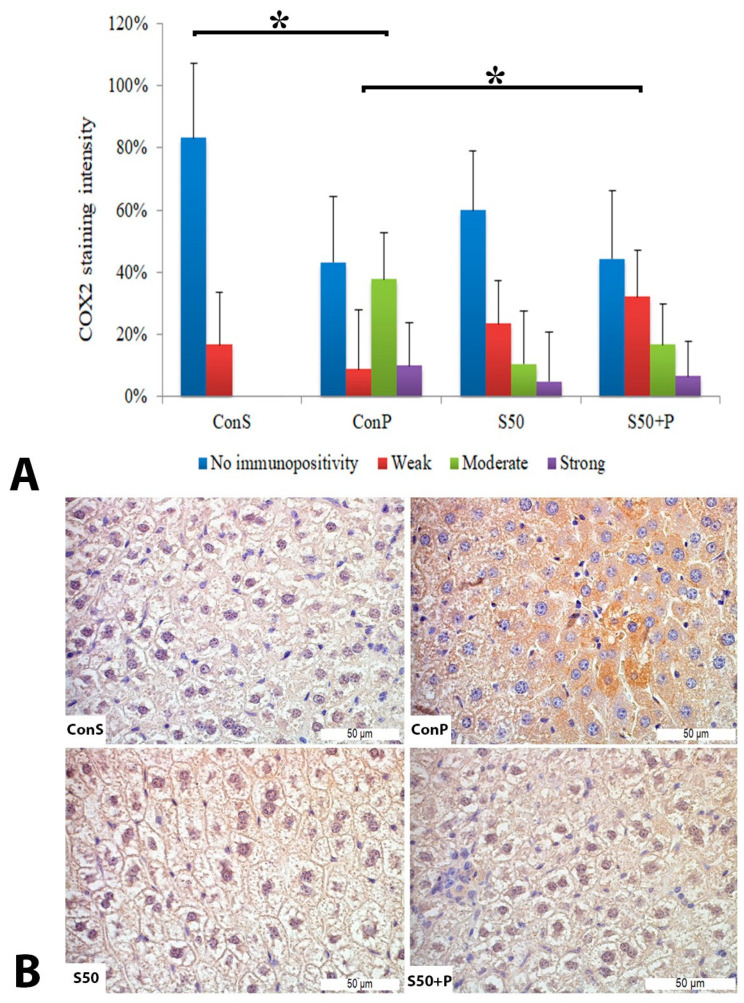
COX2 expression in the liver tissue: (**A**) COX2 staining intensity, (**B**) histopathological analysis of COX2 staining in the liver tissue: Con S (400×); Con P (400×); S50 (400×); S50+P (400×); * a statistical significance level of *p* < 0.05 between the groups indicated by the ends of the square brackets.

**Figure 3 pharmaceutics-16-00520-f003:**
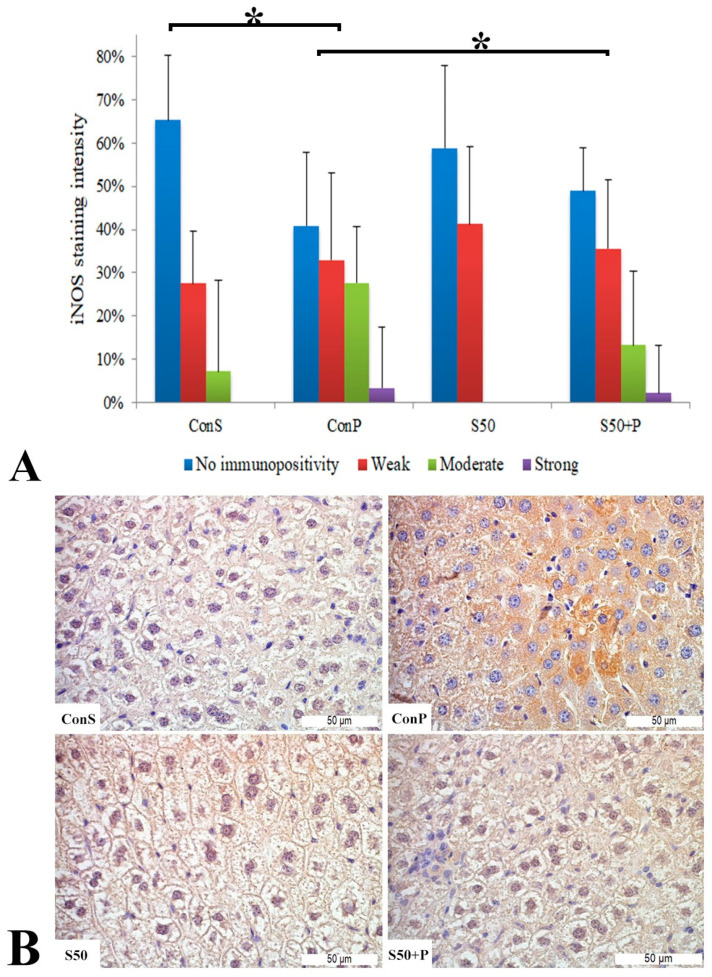
iNOS expression in the liver tissue: (**A**) iNOS staining intensity, (**B**) histopathological analysis of iNOS staining in the liver tissue: Con S (400×); Con P (400×); S50 (400×); S50+P (400×); * a statistical significance level of *p* < 0.05 between the groups indicated by the ends of the square brackets.

**Figure 4 pharmaceutics-16-00520-f004:**
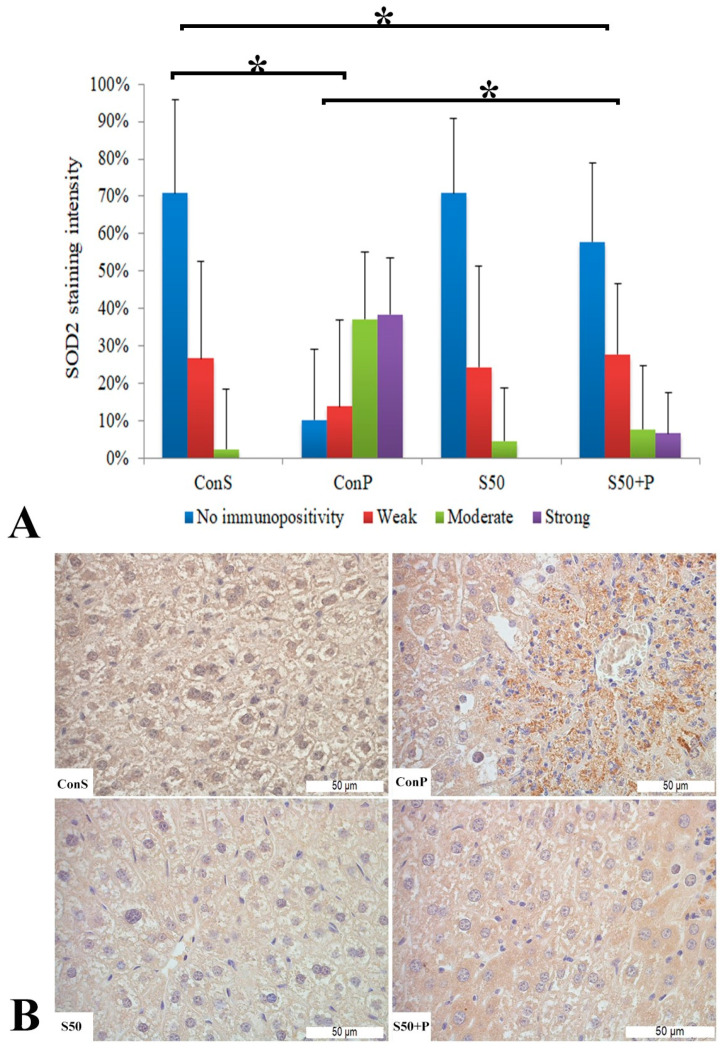
SOD2 expression in the liver tissue: (**A**) SOD2 staining intensity, (**B**) histopathological analysis of SOD2 staining in the liver tissue: Con S (400×); Con P (400×); S50 (400×); S50+P (400×); * a statistical significance level of *p* < 0.05 between the groups indicated by the ends of the square brackets.

**Figure 5 pharmaceutics-16-00520-f005:**
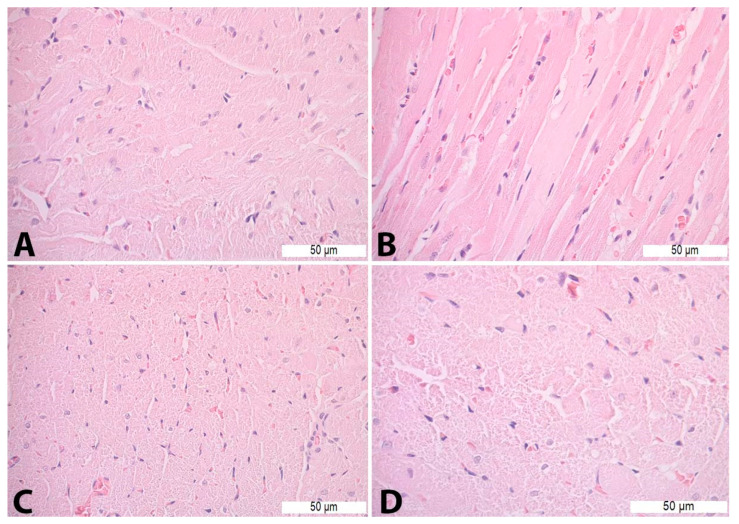
Heart tissue morphology alterations in Con P group—H&E staining: (**A**)—cardiomyocyte disorganization (400×); (**B**)—necrosis (400×); (**C**)—nucleus appearance changes (400×); and (**D**)—vacuoles (630×).

**Figure 6 pharmaceutics-16-00520-f006:**
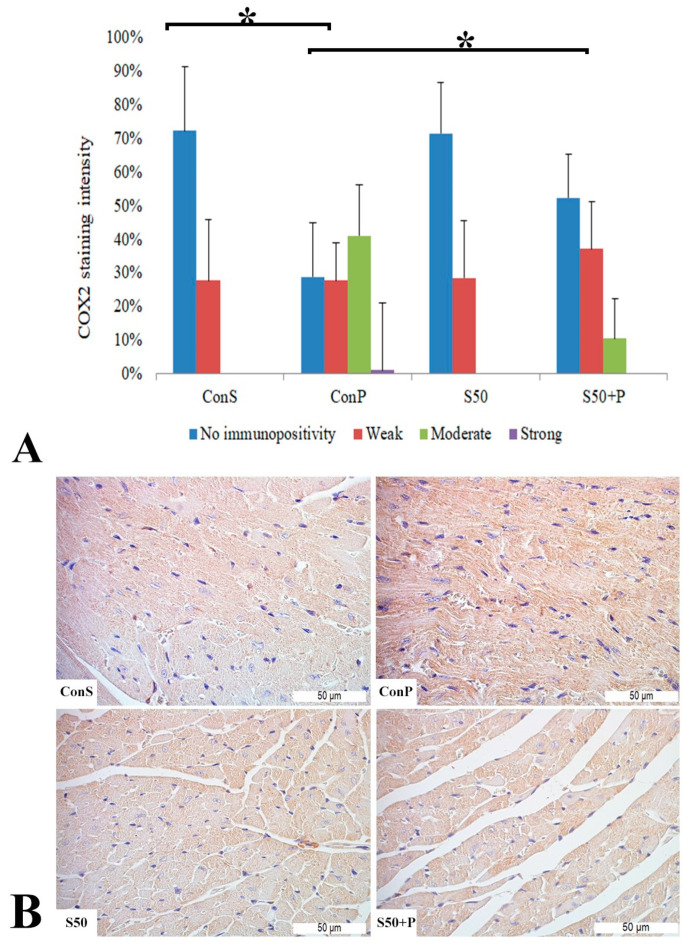
COX2 expression in the heart tissue: (**A**) COX2 staining intensity, (**B**) histopathological analysis of COX2 staining in the heart: Con S (400×); Con P (400×); S50 (400×); S50+P (400×); * a statistical significance level of *p* < 0.05 between the groups indicated by the ends of the square brackets.

**Figure 7 pharmaceutics-16-00520-f007:**
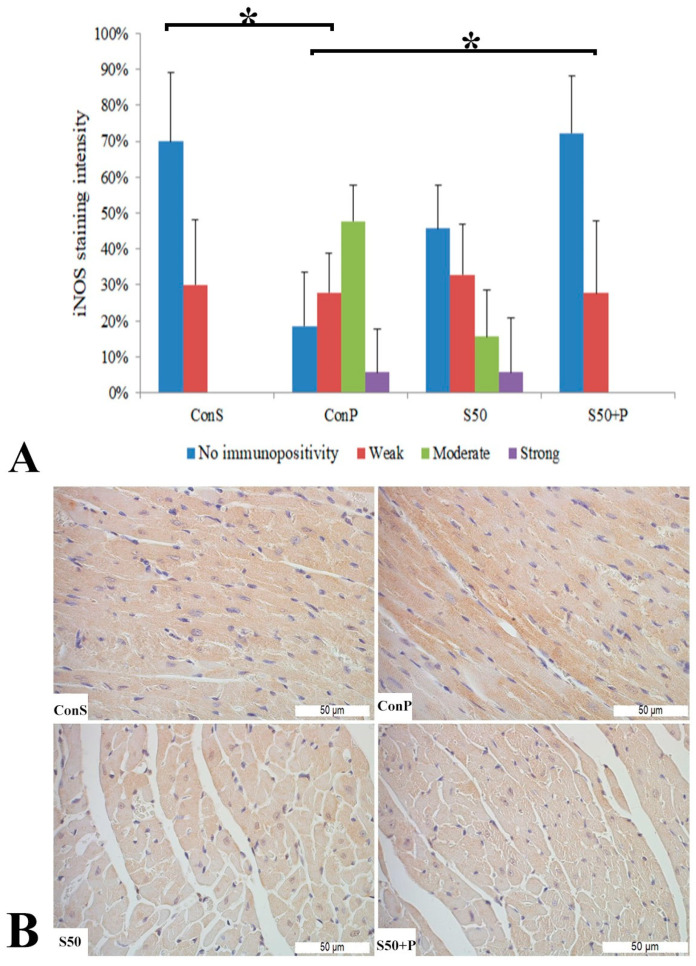
iNOS expression in the heart tissue: (**A**) iNOS staining intensity, (**B**) histopathological analysis of iNOS staining in the heart—iNOS staining: Con S (400×); Con P (400×); S50 (400×); S50+P (400×); * a statistical significance level of *p* <0.05 between the groups indicated by the ends of the square brackets.

**Figure 8 pharmaceutics-16-00520-f008:**
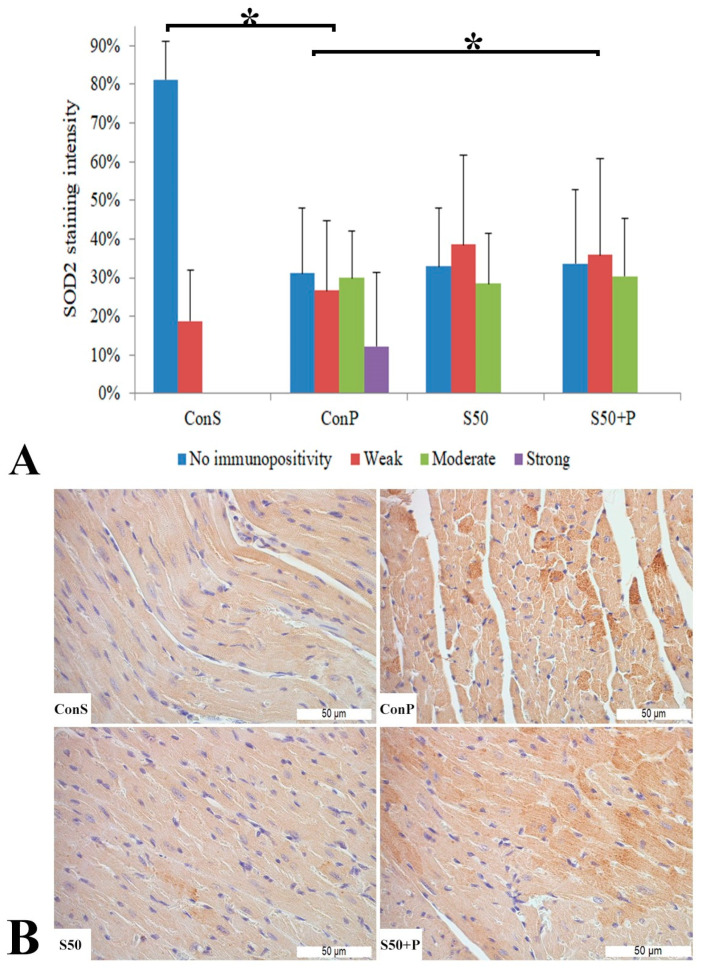
SOD2 expression in the heart tissue: (**A**) SOD2 staining intensity, (**B**) histopathological analysis of SOD2 staining in the heart: Con S (400×); Con P (400×); S50 (400×); S50+P (400×); * a statistical significance level of *p* < 0.05 between the groups indicated by the ends of the square brackets.

**Table 1 pharmaceutics-16-00520-t001:** In vitro oxidative stress tests.

	DPPH (mM TE/g)	FRAP (mM Fe^2+^/g)	ABTS (mM TE/g)
Silymarin	0.7148 ± 0.0064	4.6759 ± 0.0208	5.0478 ± 0.0283

**Table 2 pharmaceutics-16-00520-t002:** Liver function tests, kidney function tests, and lipid profile according to the animal groups.

	Con S	Con P	S50	S50+P
ALT (U/L)	49.83 ± 5.34	72.75 ± 12.45 ^a^	53.80 ± 6.01 ^b^	56.00 ± 8.29 ^b^
AST (U/L)	127.80 ± 33.34	257.00 ± 46.94 ^a^	147.50 ± 13.82 ^b^	151.29 ± 39.38 ^b^
Total bilirubin (µmol/L)	2.06 ± 0.25	2.50 ± 0.54	2.30 ± 0.24	2.62 ± 0.28
Urea (mmol/L)	6.50 ± 0.45	7.68 ± 1.03	6.01 ± 0.17	6.70 ± 0.28
Creatinine (μmol/L)	29.78 ± 1.08	33.04 ± 5.30	29.64 ± 0.85	31.44 ± 1.95
Uric acid (μmol/L)	265.00 ± 5.48	256.67 ± 27.33	218.33 ± 19.41	180.00 ± 17.32 ^a,b^
Triglycerides (mmol/L)	1.31 ± 0.23	2.25 ± 0.12 ^a^	1.27 ± 0.10 ^b^	1.69 ± 0.45 ^b^
Cholesterol (mmol/L)	3.81 ± 0.36	3.93 ± 0.99	3.51 ± 0.27	3.85 ± 0.33
HDL (mmol/L)	2.11 ± 0.14	1.88 ± 0.48	2.01 ± 0.13	1.98 ± 0.29
LDL (mmol/L)	0.98 ± 0.29	1.25 ± 0.66	0.92 ± 0.15	1.12 ± 0.22

^a^ *p* < 0.05 Significantly different from Con S group. ^b^ *p* < 0.05 Significantly different from Con P group.

**Table 3 pharmaceutics-16-00520-t003:** Effects of silymarin on paracetamol-induced oxidative stress—liver tissue.

	Con S	Con P	S50	S50+P
LP (nmol MDA/mg proteins)	0.116 ± 0.021	0.177 ± 0.036 ^a^	0.092 ± 0.023	0.131 ± 0.014 ^b^
SOD (U/mg proteins)	26.25 ± 1.91	13.57 ± 2.81 ^a^	27.22 ± 3.70	20.89 ± 1.82 ^b^
CAT (U/mg proteins)	82.03 ± 9.46	46.19 ± 4.50 ^a^	80.41 ± 5.16	75.72 ± 3.78 ^b^
GPx (nmol/mg proteins)	45.94 ± 3.93	29.87 ± 4.59 ^a^	46.10 ± 2.74 ^b^	38.99 ± 4.60 ^b^
GR (nmol/mg proteins)	16.85 ± 3.55	13.40 ± 3.23	14.26 ± 2.69	15.48 ± 3.65
GST (nmol/mg proteins)	40.52 ± 4.94	17.89 ± 2.97 ^a^	38.47 ± 5.34 ^a,b^	31.93 ± 4.38 ^b^

^a^ *p* < 0.05 Significantly different from Con S group. ^b^ *p* < 0.05 Significantly different from Con P group.

**Table 4 pharmaceutics-16-00520-t004:** Effects of silymarin on paracetamol-induced oxidative stress—heart tissue.

	Con S	Con P	S50	S50+P
LP (nmol MDA/mg proteins)	0.82 ± 0.010	0.138 ± 0.019 ^a^	0.076 ± 0.017 ^b^	0.098 ± 0.012 ^b^
SOD (U/mg proteins)	16.47 ± 1.73	11.25 ± 1.80 ^a^	15.57 ± 1.51	15.12 ± 1.61 ^b^
CAT (U/mg proteins)	56.19 ± 4.50	44.03 ± 3.72 ^a^	53.38 ± 4.38	51.74 ± 2.87 ^b^
GPx (nmol/mg proteins)	39.87 ± 4.59	36.05 ± 3.93	43.10 ± 5.74	38.99 ± 4.60
GR (nmol/mg proteins)	13.40 ± 2.01	10.85 ± 2.49	14.48 ± 3.65	12.26 ± 2.69
GST (nmol/mg proteins)	27.89 ± 2.97	23.52 ± 1.94	29.47 ± 2.34	25.93 ± 1.38

^a^ *p* < 0.05 Significantly different from Con S group. ^b^ *p* < 0.05 Significantly different from Con P group.

**Table 5 pharmaceutics-16-00520-t005:** Presence of necrosis and inflammatory infiltrate in the liver tissue.

	Con S	Con P	S50	S50+P
Necrosis	0%	7.8% ^a^	0%	3.3% ^b^
Inflamatory infiltrate	0%	15.9% ^a^	1.2%	8.2% ^b^

^a^ *p* < 0.05 Significantly different from Con S group. ^b^ *p* < 0.05 Significantly different from Con P group.

**Table 6 pharmaceutics-16-00520-t006:** Percentage of area in the heart tissue stained by COX2, iNOS, and SOD2.

	Con S	Con P	S50	S50+P
COX2	2%	19.7% ^a^	5.4%	10.58% ^b^
iNOS	4.5%	27.3% ^a^	16.78%	5.94% ^b^
SOD2	1.4%	15.2% ^a^	10.6%	8.8% ^b^

^a^ *p* < 0.05 Significantly different from Con S group. ^b^ *p* < 0.05 Significantly different from Con P group.

## Data Availability

The data presented in this study are contained within the article.

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
