# Peer review of "Cardioprotective and Hepatoprotective Potential of Silymarin in Paracetamol-Induced Oxidative Stress"

_pharmaceutics, 2024, doi:10.3390/pharmaceutics16040520_

Round 1
Reviewer 1 Report
Comments and Suggestions for Authors
1. Please standardise the nomenclature throughout the manuscript (e.g. sylimarin/silimarin etc).
2. Please write the names of compounds and ions correctly, numbers in subscripts or superscripts.
3. At what concentration was the DPPH solution used for the determinations prepared?
4. In what concentration range was silymarin tested for antioxidant activity and vitro?
5. I have doubts whether the use of Trolox for the standard curve is correct here. I would rather suggest in these three systems to present the data in a graph (e.g. changes in DPPH absorbance on the Y-axis and sillimarol concentration on the X-axis) and determine the IC50 parameter.
6. The in vitro antioxidant results were too poorly described and discussed.
7. I suggest additionally determining ROS levels in in vivo systems.
8. please refine all graphs presented. Describe the Y-axis on all graphs, add standard deviations on all bars and statistical significance. Please remove the incomplete box surrounding the graphs.
Author Response
Dear Reviewer,
Many thanks for the positive feedback. I am attaching our responses to the reviewers' comments. Also, all changes made in the manuscript are highlighted and can be easily followed. We hope that the revised paper will meet all demands related to good quality and scientific contribution, required for publication in Pharmaceutics, as well as that further research in this area will be interesting and significant for readers.
Reviewers' comments
- Please standardise the nomenclature throughout the manuscript (e.g. sylimarin/silimarin etc).
Response: nomenclature in the text is standardized.
- Please write the names of compounds and ions correctly, numbers in subscripts or superscripts.
Response: names of compounds and ions, numbers in subscripts or superscripts are written correctly.
- At what concentration was the DPPH solution used for the determinations prepared?
Response: The DPPH methanolic solution was previously prepared in concentration of 26 mg/L. Section 2.4.1. was adjusted accordingly.
- In what concentration range was silymarin tested for antioxidant activity and vitro?
Response: Concentration range for the silymarin was evaluated at following concentrations of stock silymarin solution:
- DPPH test: 0.1-1 mg/mL
- FRAP test: 0.1-0.5 mg/mL
- ABTS test: 0.1-0.5 mg/mL
- I have doubts whether the use of Trolox for the standard curve is correct here. I would rather suggest in these three systems to present the data in a graph (e.g. changes in DPPH absorbance on the Y-axis and sillimarol concentration on the X-axis) and determine the IC50 parameter.
Response: The authors agree with reviewer’s comment that use of Trolox for the standard curve is not the best solution. However, we have chosen this approach based on two premises: 1) Our systems are calibrated using these standards and we previously had good experience by expressing the results in this way in order to make them more comparable with other scientific results and 2) recent study used the same units for expressing the antioxidant activity of Silybum marianum (Villegas-Aguilar, M. D. C., Sánchez-Marzo, N., Fernández-Ochoa, Á., Del Río, C., Montaner, J., Micol, V., ... & Segura-Carretero, A. (2024). Evaluation of Bioactive Effects of Five Plant Extracts with Different Phenolic Compositions against Different Therapeutic Targets. Antioxidants, 13(2), 217.). Therefore, we expressed date in this way and compared them in discussion.
- The in vitro antioxidant results were too poorly described and discussed.
Response: The authors improved this section according to reviewer’s remark.
- I suggest additionally determining ROS levels in in vivo systems.
Response: ROS levels in in vivo systems are additionally determined in both, liver and hearth tissue.
- please refine all graphs presented. Describe the Y-axis on all graphs, add standard deviations on all bars and statistical significance. Please remove the incomplete box surrounding the graphs.
Response: All graphs are corrected – Y-axis is now labeled, standard deviations and statistical significances are added.
Once more, many thanks to you for the precious time and efforts spent on our manuscript. As a result of the valuable comments and suggestions, we believe we have been able to improve the quality of our manuscript. We sincerely hope our manuscript is now suitable for publication in Pharmaceutics, and look forward to hearing from you at your earliest convenience.
Reviewer 2 Report
Comments and Suggestions for Authors
The manuscript entitled “Cardioprotective and hepatoprotective potential of silymarin in paracetamol induced oxidative stress” evaluated the potential cardioprotective and hepatoprotective effects of the extract silymarin against the oxidative stress induced by paracetamol in male Swiss Webster mice.
The manuscript fits with the aims of Pharmaceutics, however, in the present form, the overall presentation is not adequately organized, the obtained results are not appropriately presented, and the presence of many typographical and grammatical errors greatly affects the quality of the manuscript. Therefore, several integrations should be carried out to improve the manuscript.
Specific comments:
1) Many grammatical and typographical mistakes are present in the manuscript. Extensive editing of style is required.
For example:
- Page 1, lines 21, 40, and 43; page 2, line 46: the name of the herb Silybum marianum and the mushroom Amanita phalloides should be italicized.
- Page 1, line 30: please correct “significant” to “significant”.
- The correct use of superscript/subscript should be checked in the text (for example: page 2, line 70; page 4, lines 161, 163, 167, and 168).
2) Introduction: The aim of the study should be clearly stated, with an indication of the most important items/objectives of the research. More literature references should be included in the Introduction to justify the potential therapeutic application of silymarin in paracetamol-induced liver and heart injury in rodents.
3) The specific phenol composition of silymarin used in this study (from Sigma) should be reported in the manuscript.
4) Several chemicals indicated in the manuscript, such as 2,2-diphenyl-1-picrylhydrazyl, 2,4,6-tris (2-pyridyl)-s triazine, Fe2SO4 and others, should be indicated in Paragraph 2.1 Chemicals.
5) There is confusion in data presentation and all reported Graphs/Figures should be completely reorganized. All graphs and images should be titled as Figures. For example, Graph 1/Figure 2 should be combined in a single Figure (Figure 2). The same consideration is valid for Graph 2/Figure 3, Graph 3/Figure 4, Graph 4/Figure 6, Graph 5/Figure 7, and Graph 6/Figure 8. Therefore, the reorganization of the captions to Figures and the text of the “Results” paragraph is necessary.
6) No standard deviations are indicated for data reported in each Graph. Please add standard deviations in all Graphs.
7) The decimal comma should be replaced with a point in the numbers reported in all the text of the manuscript, including Table 1, Table 2, Table 3, and Table 4.
Comments on the Quality of English Language
Many grammatical and typographical mistakes are present in the manuscript.
Extensive editing of style is required.
Author Response
Dear Reviewer,
Many thanks for the positive feedback. I am attaching our responses to the reviewers' comments. Also, all changes made in the manuscript are highlighted and can be easily followed. We hope that the revised paper will meet all demands related to good quality and scientific contribution, required for publication in Pharmaceutics, as well as that further research in this area will be interesting and significant for readers.
Reviewers' comments
The manuscript entitled “Cardioprotective and hepatoprotective potential of silymarin in paracetamol induced oxidative stress” evaluated the potential cardioprotective and hepatoprotective effects of the extract silymarin against the oxidative stress induced by paracetamol in male Swiss Webster mice.
The manuscript fits with the aims of Pharmaceutics, however, in the present form, the overall presentation is not adequately organized, the obtained results are not appropriately presented, and the presence of many typographical and grammatical errors greatly affects the quality of the manuscript. Therefore, several integrations should be carried out to improve the manuscript.
Specific comments:
1) Many grammatical and typographical mistakes are present in the manuscript. Extensive editing of style is required.
For example:
- Page 1, lines 21, 40, and 43; page 2, line 46: the name of the herb Silybum marianum and the mushroom Amanita phalloides should be italicized.
- Page 1, line 30: please correct “significant” to “significant”.
- The correct use of superscript/subscript should be checked in the text (for example: page 2, line 70; page 4, lines 161, 163, 167, and 168).
Response: typographical and grammatical errors have been corrected and extensive editing of style was performed. The use of superscript/subscript was checked and corrected in the manuscript.
2) Introduction: The aim of the study should be clearly stated, with an indication of the most important items/objectives of the research. More literature references should be included in the Introduction to justify the potential therapeutic application of silymarin in paracetamol-induced liver and heart injury in rodents.
Response: the aim is more clearly stated and four more references were added to Introduction.
3) The specific phenol composition of silymarin used in this study (from Sigma) should be reported in the manuscript.
Response: Sigma’s phenol composition of silymarin used in this study is reported in the manuscript, Paragraph 2.1
4) Several chemicals indicated in the manuscript, such as 2,2-diphenyl-1-picrylhydrazyl, 2,4,6-tris (2-pyridyl)-s triazine, Fe2SO4 and others, should be indicated in Paragraph 2.1 Chemicals.
Response: chemicals used in in vitro assays have been added to the list of chemicals in Paragraph 2.1 of the manuscript.
5) There is confusion in data presentation and all reported Graphs/Figures should be completely reorganized. All graphs and images should be titled as Figures. For example, Graph 1/Figure 2 should be combined in a single Figure (Figure 2). The same consideration is valid for Graph 2/Figure 3, Graph 3/Figure 4, Graph 4/Figure 6, Graph 5/Figure 7, and Graph 6/Figure 8. Therefore, the reorganization of the captions to Figures and the text of the “Results” paragraph is necessary.
Response: Graph and figure representing the same antibody used for tissue analysis are incorporated in one plate, titled Figure, numbered as follows:
Previous version: Current version:
Figure 1 Figure 1
Graph 1/Figure 2 Figure 2
Graph 3/Figure 4 Figure 4
Figure 5 Figure 5
Graph 4/Figure 6 Figure 6
Graph 5/Figure 7 Figure 7
Graph 6/Figure 8 Figure 8
6) No standard deviations are indicated for data reported in each Graph. Please add standard deviations in all Graphs.
Response: Standard deviations are indicated in each Graph incorporated in the Figures 2-4, and Figures 6-8.
7) The decimal comma should be replaced with a point in the numbers reported in all the text of the manuscript, including Table 1, Table 2, Table 3, and Table 4.
Response: decimal commas are replaced with a point in the numbers reported in all the text of the manuscript, including Table 1, Table 2, Table 3, and Table 4.
Once more, many thanks to you for the precious time and efforts spent on our manuscript. As a result of the valuable comments and suggestions, we believe we have been able to improve the quality of our manuscript. We sincerely hope our manuscript is now suitable for publication in Pharmaceutics, and look forward to hearing from you at your earliest convenience.
Round 2
Reviewer 2 Report
Comments and Suggestions for Authors
All the suggested corrections have been made in the revised version of the manuscript.
Figures are not adequately organized.
The separate parts in Figures assembled from multiple photographs/images/graphs should be indicated with letters (A, B, C, ..) instead of numbers.
Author Response
Thank you for your kind suggestion, the separate parts in all Figures assembled from multiple photographs/images/graphs are now indicated with letters (A, B, C, ..) instead of numbers. If some additional corrections are needed on the Figures, please give us specific instructions.